# Lavender–Neroli Aromatherapy for Reducing Dental Anxiety and Pain in Children During Anesthesia: A Two-Arm Randomized Controlled Trial

**DOI:** 10.3390/medsci13030166

**Published:** 2025-09-01

**Authors:** Rama Abdalhai, Yasser Alsayed Tolibah, Racha Alkhatib, Chaza Kouchaji, Ziad D. Baghdadi

**Affiliations:** 1Department of Pediatric Dentistry, Faculty of Dentistry, Damascus University, Damascus P.O. Box 3062, Syria; ramaabdulhai77@gmai.com (R.A.); yasseralsayedtolibah@gmail.com (Y.A.T.); shazako@yahoo.com (C.K.); 2Department of Pharmacognosy, Faculty of Pharmacology, Damascus University, Damascus P.O. Box 3062, Syria; khatibracha@hotmail.com; 3Department of Preventive Dental Sciences, Division of Pediatric Dentistry, University of Manitoba, Winnipeg, MB R3E 0W2, Canada; 4Dr. Gerald Niznick College of Dentistry, University of Manitoba, P131B, 780 Bannatyne Avenue, Winnipeg, MB R3Y 1P5, Canada

**Keywords:** pediatric dentistry, aromatherapy, lavender–neroli oil, dental anxiety

## Abstract

**Objective.** This randomized controlled trial evaluated the efficacy of lavender–neroli oil aromatherapy in managing dental anxiety and pain in children undergoing inferior alveolar nerve block (IANB) anesthesia. **Methods.** Fifty-four children aged 6–11 years were randomly assigned to either a control group or an aromatherapy group. Children in the control group were asked to wear a regular scented-free nitrous oxide mask. Children in the control group were asked to wear a regular scented-free nitrous oxide mask. Children in the intervention group inhaled lavender–neroli oil via a nitrous oxide nasal mask for 5 min before and during IANB administration. Anxiety and pain levels were assessed pre-and post-treatment using the Facial Image Scale (FIS), Face–Legs–Activity–Cry–Consolability (FLACC) scale, and vital signs (heart rate, blood pressure, oxygen saturation). The collected data were statistically analyzed using SPSS software 20. The Mann–Whitney U test was used for analyzing FIS results, and the independent T test and T Paired test were used for analyzing heart rate, blood pressure, and oxygen saturation results. **Results.** Results demonstrated significantly lower anxiety, heart rate, blood pressure, and pain scores in the aromatherapy group compared to the control group (*p* < 0.05), with no significant change in oxygen saturation. **Conclusions.** Lavender–neroli aromatherapy is a safe, low-cost, and effective adjunct to reduce anxiety and discomfort during pediatric dental anesthesia.

## 1. Introduction

Dental anxiety is a prevalent and persistent issue among children, often hindering their ability to receive appropriate dental care. With reported prevalence ranging from 13% to 23.9% [1,2], anxiety significantly impacts children’s willingness to undergo dental procedures [3]. Triggers include the visual and tactile presence of needles, high-pitched sounds of dental handpieces, and distinctive odors associated with dental materials [4,5].

Children with elevated dental fear are more likely to present with untreated caries and compromised oral health due to avoidance behaviors. This emphasizes the necessity of managing pediatric dental anxiety to ensure successful treatment outcomes [6].

Many pharmacological and non-pharmacological methods have been used to elevate dental anxiety. Some parents reject pharmacological techniques, including nitrous oxide and general anesthesia, due to the risks associated with them in comparison to the importance of primary teeth maintenance and the need for specialized equipment [7]. Moreover, with pharmacological methods, even with the presence of specialists, life-threatening risks can occur, such as loss of protective airway reflex, cardiovascular instability, and respiratory depression [8].

Recently, the effectiveness of aromatherapy as one of the complementary alternative medicines has been suggested for reducing anxiety and pain during medical procedures [9,10,11].

Aromatherapy seeks to achieve therapeutic benefits such as increased calmness, improved physical well-being, and a reduction in anxiety and pain. It relies on essential oils extracted from raw plant materials, which are defined as the cultivated plant parts that naturally produce essential oils. These parts may include flowers, leaves, peels, fruits, bark, or seeds [12,13]. Furthermore, aromatherapy has garnered attention for its potential to reduce anxiety, attributed to its perceived safety profile with minimal side effects, ease of application, and affordability compared to synthetic drugs [14].

Essential oils are a mixture of different compounds that enter the body by inhalation, skin massages, baths, and orally to obtain analgesic, anxiolytic, sedative, and anti-inflammatory effects [15]. Inhalation is the most common way associated with aromatherapy and affects the sense of smell [16], as the odor from essential oils can affect the limbic system, where emotions develop in the brain, including the amygdala and hippocampus, and produce anxiolytic effects [17].

Lavender oil extract from Lavandula angustifolia flowers is one of the most important essential oils used in aromatherapy, gaining popularity due to its anxiolytic, analgesic, and sedative properties [18]. It belongs to the Lamiaceae family with sedative activity due to its constituents of linalool and linalyl acetate [19]. Neroli is an essential oil extracted by hydrodistillation from citrus aurantium or bitter orange flowers, which belong to the Rutaceae family [20]. It is widely used in aromatherapy due to its effect on the central nervous system, including sedative, analgesic, and anti-inflammatory properties [21].

Recently, several studies have shown the effectiveness of contemporary alternative approaches, such as aromatherapy with multiple essential oils, in reducing anxiety and pain during needle insertion in healthy volunteers, hemodialyzed patients, and children [9,10,11]. Moreover, the effect of aromatherapy in reducing dental anxiety has been studied in waiting rooms [22,23]. As for the pediatric patient, it has been studied during prophylaxis and invasive dental treatment [23,24,25,26,27], where these studies mostly used lavender or sweet orange oils only. However, no study has focused on the effectiveness of an essential oil mixture on children’s anxiety during anesthesia injection. In a previous study, lavender with neroli oil was used as a blend in combination with music therapy during IANB injection in children [28]. This study aimed to investigate the efficacy of a lavender–neroli oil mixture alone, without other factors, in reducing dental anxiety, vital signs, and pain during inferior alveolar nerve block (IANB) anesthesia injections in children. The investigation sought to isolate the effect of the aromatherapy intervention and determine whether it could serve as a non-invasive, supportive measure during pediatric dental procedures. The null hypothesis (H_0_) is that the use of a lavender–neroli oil mixture has no effect on reducing dental anxiety, physiological responses (vital signs), or perceived pain during IANB administration in children compared to no aromatherapy intervention.

## 2. Materials and Methods

### 2.1. Study Design, Settings, and Ethical Approval

This single-blinded, randomized clinical trial employed a two-arm, parallel superiority design with a 1:1 allocation ratio, conducted from October 2021 to January 2023 at the Department of Pediatric Dentistry at the Faculty of Dentistry, Damascus University, Syria. This study adhered to the ethical guidelines of the Declaration of Helsinki and its later amendments and received ethical approval from the Local Research Ethics Committee of the Faculty of Dentistry (Approval No. UDDS-502-220420121/SRC-3183). The project was registered at the clinical trials government with an identifier (NCT05759286) on 3 August 2023.

### 2.2. Sample Size Calculation

The sample size was estimated using G* Power 3.1.9.4 (Heinrich-Heine-Universität, Düsseldorf, Germany) based on the changes in pulse rate values. A minimum total sample size of 54 patients (27 in each group) was found to be sufficient for a level of significance of 0.05, a power of 85%, and an effect size estimated from previous research [29].

### 2.3. Recruitment and Eligibility Criteria

Two hundred and fifteen children aged between 6 and 11 years were referred to the Department of Pediatric Dentistry during the study period for a dental procedure in the mandibular jaw. The children were investigated by the principal investigator (R.Ab.), who searched for healthy children who had sufficient cognitive skills to complete the self-report scale, who needed dental treatment recommended anesthesia with alveolar nerve block injection, and who recorded grade 2 on the Frankel behavior scale (positive). One hundred and fifty children met these inclusion criteria. Children were excluded if parents refused to participate in this study, if children had asthma, a cold, or any respiratory hypersensitivity, if children took analgesics or nonsteroidal anti-inflammatory Drugs (NSAIDs) in the last 8 h, if children had a mental or physical disability, or if children were allergic to the oils used in this study. Finally, fifty-four children were included in the current study. All included children whose parents agreed to participate in the study signed an informed consent sheet after hearing all of the details about the trial and its therapeutic nature.

### 2.4. Randomization

Children were assigned to the control group or the aromatherapy group using the simple randomization method at an allocation ratio of 1:1, and a random sequence was created by using the website www.random.org. accessed on 1 May 2023.

Thus, children were assigned to two groups: Group 1, the control group (*n* = 27), and Group 2, the aromatherapy group (*n* = 27).

### 2.5. Blinding

As the current study was an interventional study, the treating clinician was aware of whether the child had undergone aromatherapy or not due to the aromatic spread of the essential oils in the treatment room. Moreover, the children involved in this study were not blinded. Although children in both groups wear an aromatherapy mask, they will be aware if they smell essential oils or not. The assessment of treatment outcomes was conducted by two trained Ph.D. student researchers, who were calibrated to the evaluation criteria and blinded to the type of therapy used. Both groups of children wore a modified nitrous mask. The assessment was conducted by viewing a video recorded from a camera mounted over the patient’s head, which showed the vital signs meter and the patient’s body for objective pain assessment. The assessors had previously gained experience in using the FLACC scale. The inter-rater reliability for assessors was tested using the Kappa statistics, which revealed high compatibility for both assessors, 0.932 and 0.879 [30].

### 2.6. Intervention Procedures

After taking demographic information and being informed by parents, the children were randomly divided into two groups.

Control group: The child was asked to put the nasal nitrous oxide mask with the box on it, which was empty and did not include any aroma, as a placebo effect.

Aromatherapy group: First, a mixture of lavender–neroli oils was prepared by a specialist in the Faculty of Pharmacology Department of Pharmacognosy (R.Al.). Essential oils were obtained from (Biocham Natural Extract Co., Damascus, Syria). (100% lavender oil (L. Angustifolia), and 100% neroli oil (citrus aurantium).

The main components of essential oils were determined by Gas Chromatography (GC) and found to be 37% linalool, 11.6% camphor, 9.9% 1.8 cineole, 5.5% linalyl acetate in lavender oil, and 23.4% linalool, 15.5% linalyl acetate,12.3% trans-Nerolidol, 11.9% limonene, 7.7% β pinene in neroli oils, this data was provided by the delivering company. The main components of lavender oil, including linalool and linalyl acetate, were confirmed using Gas Chromatography (GC), consistent with prior literature [14].

A total of 2.3 mL of lavender oil was mixed with 0.9 mL of neroli oil, and the total mixture was diluted to 20 mL using grapeseed oil.

In the dental treatment, the child was asked to inhale the aroma of the oil mixture through a modified nasal nitrous oxide mask (Accturon, Hu-Friedy Mfg. Co., LLC, Phoenix, AZ, USA). A 3D-printed box, which was perforated from the top and bottom to allow passing the air, was applied on the circle hole of the mask; three drops of the oil mixture were poured by a disposable pipette with a 3 mm tip diameter (Sunnypack, Dallas, Australia) on three cotton balls which were put in the 3 of box cavities and leave on cavity empty to ensure continuous ventilation and not be blocked by the cotton balls (Figure 1A–C). Following its modification, the nasal mask was reframed and introduced to children not as a clinical device but as a playful object. The mask, introduced to resemble familiar figures such as a cat or bear, created an appearance that aligned more closely with a toy than with medical equipment. This design adaptation, coupled with the use of simplified, child-friendly terminology, contributed to reducing anxiety and enhancing cooperation during dental procedures. By integrating elements of play and imagination, the intervention improved children’s acceptance of care and facilitated their understanding of the procedures being performed. Such approaches underscore the value of child-centered communication and environmental adaptation in pediatric dentistry, where minimizing fear can significantly influence treatment outcomes and long-term attitudes toward oral healthcare.

Children in this group inhaled the aromatic oils for 5 min before and during anesthesia using special masks that differed from the masks in the control group. They were treated in a different room from the children of the control group, to be affected by the trace of essential oils remaining in the masks or the room of the aromatherapy group.

Children in both groups received IANB injections by the same researcher using lidocaine 2% with 1:80,000 epinephrine (Huons Lidocaine HCL, Seoul, Republic of Korea) and a 27-gauge needle (Kohope, Shanghai, China), which was used at a depth of 15 mL where 1 mL of the anesthetic’s solution was injected [31].

### 2.7. Outcomes and Measurements

Dental anxiety and pain scores were defined as primary outcome measures, whereas vital signs, blood pressure, heart rate, and oxygen saturation were considered secondary outcome measures.

A dental anxiety assessment was conducted using the Facial Image Scale (FIS). It consists of a row of 5 faces ranging from very happy to very sad, and the child points to the face that reflects his feelings. The scale scored 1 to the most positive face and 5 to the most negative face [32,33].

The Arabic version of the Face-Legs-Arms-Cry-Consolability (FLACC) scale was used to measure pain during anesthesia injection, as it is a reliable tool to assess pain in children aged between 6 and 14 years in dental treatment [34]. Five pain behaviors are included in FLACC; each category of behavior is measured on a 0–2 scale, so the whole scale scores range between 0 and 10 [2].

For vital signs, heart rate and oxygen saturation were recorded using pulse oximeters (Meditech Equipment, Qingdao, China), whereas an electronic blood pressure monitor took diastolic and systolic blood pressure with a cuff on the elbow (O_2_ medical systems, Hyderabad, India).

Vital signs were recorded before and after 1 min of anesthesia.

### 2.8. Statistical Analysis

The data of the present study were analyzed using the statistical package for the social sciences software (Version 20, IBM SPSS Inc., Chicago, IL, USA). Sample Descriptive statistics were used by number and percentage for gender and the mean and standard deviation for age.

The normality of the data distribution was determined using the Kolmogorov–Smirnov test, where data of all outcomes showed normal distribution. The confidence was determined at 95% with a level of significance of 0.05.

The differences in FIS variables between groups were analyzed using the Mann–Whitney U test, and intra-group comparisons before and after anesthesia were done using the Wilcoxon signed-rank test.

Paired *t*-test was used to analyze heart rate, diastolic and systolic blood pressure, and oxygen saturation differences at different time interventions within the group.

Differences in the group in heart rate, diastolic and systolic blood pressure, oxygen saturation, and FLACC were analyzed using an independent *t*-test.

## 3. Results

The flowchart of this study is described in Figure 2.

Fifty-four children, with a mean age of 8.2 ± 1.3 years (15 boys and 12 girls) for the control group and 8.1 ± 1.5 years (12 boys and 15 girls) for the aromatherapy group, participated in the study.

There were no significant differences in dental anxiety according to FIS (*p* = 0.916) and vital signs, including heart rate (0.825), diastolic and systolic blood pressure (*p* = 0.182 and *p* = 1.00, respectively), and O_2_ saturation (*p* = 0.563) between the control group and the aromatherapy group before anesthesia injection (Table 1).

After anesthesia, there was a significant statistical reduction in FIS score in the aromatherapy group compared to the control group (*p* = 0.001) (Table 1).

For vital signs, heart rate, diastolic and systolic blood pressure were significantly reduced in the aromatherapy group compared to the control (*p* = 0.000) (*p* = 0.013) (*p* = 0.038), and no significant differences were shown in O_2_ saturation between the two groups (*p* = 0.744) (Table 1).

FLACC in the aromatherapy group showed significantly lower pain perception compared to the control after anesthesia injection (*p* = 0.042) (Table 1).

Within-group comparison (Table 2) reveals no significant differences in FIS between before and after anesthesia in the aromatherapy group (*p* = 0.600). In contrast, the control group showed a significant increase after anesthesia injection compared to baseline (*p* = 0.000) (Table 3).

Heart rate and diastolic blood pressure significantly decreased in the aromatherapy group following anesthesia injection (*p* = 0.043) (*p* = 0.011). In contrast, the control group showed a statistically significant increase in all vital signs except O_2_ saturation after anesthesia injection (Table 2) (Figure 3 and Figure 4).

## 4. Discussion

Anesthesia injections, especially INAB injections, are considered the most invasive procedures that provoke anxiety and pain during dental treatment in children [35].

There has been recent progress in non-pharmacological methods used to reduce pediatric dental anxiety, including distraction techniques such as virtual reality glasses and audiovisual tools, cognitive-behavioral strategies like stress balls, spinners, or kaleidoscopes, and relaxation techniques such as music therapy, flower therapy, aromatherapy, or acupressure. These approaches are considered adequate but primarily target the psychological dimension of anxiety [35,36,37,38,39].

Within this evolving field, a key strength of the present study lies in its novel contribution to aromatherapy—not through the use of a single essential oil, but by introducing the innovative combination of lavender and neroli oils for inhalation. This method, previously unexplored, may produce unique synergistic effects. Furthermore, the study introduced refinements in the application of aromatherapy, with particular emphasis on the safe use of essential oils—an area seldom addressed in prior research. The verification of the oils’ chemical composition via gas chromatography further enhances the credibility and reproducibility of the findings. By addressing gaps in earlier studies, this research advances the development of safe and evidence-based aromatherapy strategies for managing dental anxiety in children. The findings of this study revealed the effectiveness of inhalation of an aromatic oils blend containing lavender–neroli oils in reducing dental anxiety and pain during IANB injection in children.

The choice of this oil mixture was based on a previous study [39], which demonstrated the effectiveness of aromatherapy in reducing stress and enhancing sleep in children with skin burns. Additionally, two blends with different concentrations were used to select the optimal blend. Twenty children were asked to choose which blend they preferred.

The time frame, 5 min before the intervention, depends on the hydrophobic and lipophilic nature of the essential oils, which can help with their quick effect, as they can pass through the blood–brain barrier (BBB) quicker, which makes their effect faster and immediate [20,40]. Additionally, prolonged exposure to the molecules of essential oils for more than 20 min can confuse the sense of smell, which reduces the effects of aromatherapy [41].

Anxiety is a subjective feeling that differs from one person to another [37]. In this study, the FIS, a self-report anxiety scale, was used to determine the effect of aromatherapy on anxiety for ease of conduct in children [36,37].

Anxiety scores were statistically lower in the aromatherapy group compared to the control.

Aromatic oils have a physiological effect on the nervous system and a psychological effect on the sense of smell, which contributes to their calming effect. Moreover, it is widely believed that fragrances have the power to alter the emotional state of humans [25]. It is worth noting that the odor from oil molecules stimulates the limbic system, which controls memories and feelings and, in turn, releases neurotransmitters that promote relaxation and happiness, such as morphine, serotonin, and epinephrine [15,20].

Furthermore, the components in essential oils play an important role in their anxiolytic effects. Gas chromatography showed that the main component of lavender oil was linalool and linalyl acetate, whereas neroli consists mainly of linalool, linalyl acetate, and limonene. Linalool has sedative and anxiolytic effects by acting like benzodiazepine receptors and binding with α-amino butyric acid receptors (GABA) [42]. Additionally, Linalyl acetate has a necrotic effect on the patient’s behavior and acts as a sedative [16,17].

Moreover, neroli oil has calming effects by suppressing the inflammatory mechanism of anxiety [14]. Limonene decreased neuroinflammation factors such as Nitric oxide (NO), cyclooxygenase-2 2 (COX-2), interleukin-6 (IL-6), interleukin-1β (IL-1β), and tumor necrosis factor α (TNF-α) [43].

There is an interchangeable relation between pain and anxiety; fearful children can increase the sense of pain and cause hyperalgesia [17]. In this study, FLACC scores in the aromatherapy group showed a significant decrease in pain perception during IANB injection in children compared to the control group. This result could be due to the anesthetic properties of linalool, which are found in neroli and lavender oils. Linalool affects the somatic sensory system by selectively inhibiting the Na^+^ ion channel, producing analgesic properties [14].

Pain and anxiety trigger the Sympathetic tone of the autonomic nervous system, which increases physiological markers such as heart rate, blood pressure, and changes in respiratory rate [44].

In this study, the aromatherapy group showed a decrease in heart rate and diastolic blood pressure, with no change in systolic blood pressure, after anesthesia. In contrast, the control group experienced an increase in all vital signs, including heart rate, systolic, and diastolic blood pressure, following anesthesia injection.

Lavender oil increases Parasympathetic activity, which contrasts with the work of the Sympathetic nervous system and reduces the physiological parameters that induce more relaxation and calm [45,46].

Compared to the control group, all vital signs were decreased in the aromatherapy group except O_2_ levels, which had no significant change in the aromatherapy and control groups. This could be because the children in both groups inhale through a nasal mask, which affects saturation levels. Wearing a mask increases the difficulty of breathing due to moisture and CO_2_ levels in the dead space of the mask [41].

The results of this study follow the previous studies [17,27,28,31], which found the effectiveness of aromatherapy in reducing anxiety and pain during invasive dental procedures, including anesthesia injection.

In a previous study [32], music was used in combination with the lavender–neroli oil blend aromatherapy during IANB injection, as music can reduce anxiety by releasing dopamine and opioids from the nucleus accumbens and other parts of the limbic system, thereby stimulating the reward mechanism in the brain [47]. The results of this study revealed the effectiveness of aromatherapy with music in managing dental anxiety, but in contrast with this study, the combination of music with aromatherapy was not effective in reducing pain during anesthesia. Although citrus aroma has relaxation properties, it can also play a stimulation role [48]. Music can also alter the brainstem reflex and increase attention [49], which could increase the child’s sensory pain during injection.

Arslan et al. [27] found that lavender aroma does not prevent pain during anesthesia injection, possibly because the inhalation was conducted through a med patch before treatment, thereby reducing the effectiveness of aromatherapy during the treatment. In the current study, the child was asked to inhale the aromatic oils through a nasal mask during treatment, which increased the effectiveness of aromatherapy and helped distract the child from their pain during anesthesia.

In contrast to this study, Toet and colleagues showed that neither apple nor orange aroma was effective in reducing anxiety in the waiting room of a large dental clinic [48]. The results were affected by the noise from patients entering and exiting, as well as the effect of other smells from dental materials. In another study [30], it was indicated that the use of sweet orange oil does not significantly reduce dental anxiety and circulation activity during class I GIC restoration, a non-invasive dental procedure. This is considered because it does not provoke pain and fear. Moreover, the study of Nord and colleagues revealed no significant results on the effect of lavender and ginger aromas in providing comfort to children during peri-anesthesia procedures [50]. The FLACC scale in this study was administered by children’s parents, which affected the results and made them less reliable due to the lack of training. Another reason is that the sample did not specify whether it included children with or without disabilities.

In the current study, no adverse effects were recorded, so aromatherapy can be considered a safe method to reduce dental anxiety and pain in children during invasive dental procedures. No adverse effects related to aromatherapy were observed in either group. Essential oils were diluted in grapeseed oil to minimize irritation and applied via indirect inhalation using cotton balls in a ventilated 3D-printed chamber. Children with a history of asthma or allergies were excluded to enhance safety. Both groups were closely monitored, and no respiratory distress, allergic reactions, or other adverse effects were observed.

Although aromatherapy has a natural origin, it still has some side effects, such as skin irritation and allergies after topical application due to the ingredients of essential oils [19]. Essential oils should be diluted in vegetable oils to avoid irritation for topical use [15]. Some side effects can occur when essential oils are inhaled, and this is related to allergic reactions and asthma more than the essential oils themselves [16]. In the current study, essential oils were diluted in grape seed oil, a carrier oil with no odor, to dilute them and ensure added safety, preventing them from reaching sensitive areas. Furthermore, the mixture odor was inhaled through the 3D-printed box, which is applied to the nitrous oxide nasal mask, allowing it to be easily removed and providing oxygen to the patient in emergencies if respiratory sensitivity occurs.

The strength of this study was that anxiety and pain, along with physiological markers, were recorded; another strength is that aromatherapy was inhaled directly by the patient rather than diffusing it into the air. However, this study presents several limitations that should be acknowledged. First, no control aroma was used for comparison with the lavender–neroli oil mixture, which means the observed effects may be partially attributed to the masking of clinical odors or the general distraction caused by any pleasant scent. Second, the study did not employ a split-mouth design; thus, children in the experimental and control groups were not directly matched, which may have introduced interindividual variability. The study’s sample size was determined by focusing on heart rate as the primary indicator of anxiety and fear. This approach suggests that future research can leverage the findings to refine power calculations, potentially leading to more robust and reliable studies in the field of anxiety and fear-related research.

Although aromatherapy is generally regarded as safe, it may pose potential risks—especially in children with undiagnosed bronchial hyperreactivity or allergies. Additionally, variables such as gender, age, and previous dental experiences were neither controlled nor analyzed in subgroups, which could have influenced the perception of anxiety and pain. Future studies should incorporate stratified or subgroup analyses to enhance the accuracy and relevance of the findings.

The use of non-pharmacological methods, such as music, distraction techniques, and natural products like lavender, shows promising potential for improving child behavior and satisfaction during dental treatment in pediatric dentistry. Combining pharmacological approaches with non-pharmacological methods—for example, electronic dental anesthesia/transcutaneous electrical nerve stimulation with music, nitrous oxide with lavender and music, or nitrous oxide with lavender and distraction—may represent the future of pediatric dental care [51,52,53]. These integrative strategies warrant further exploration and research.

## 5. Conclusions

The results of this study revealed the effectiveness of aromatherapy with lavender–neroli oils as a safe, low-cost, and simple non-pharmacological method for reducing dental anxiety and pain in children during dental anesthesia injection.

## Figures and Tables

**Figure 1 medsci-13-00166-f001:**
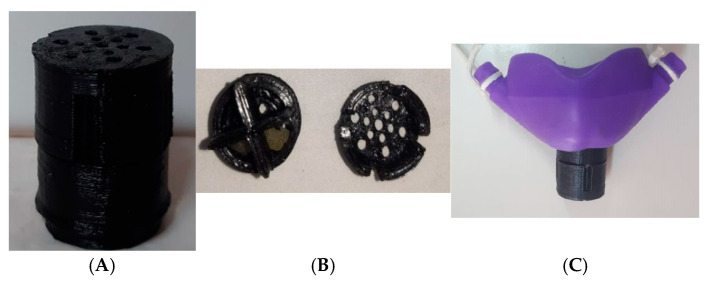
Modified nasal nitrous oxide mask; (**A**): 3D-printed box, (**B**): Top and Bottom cover of the box, and (**C**): Final shape of the box and the nasal mask.

**Figure 2 medsci-13-00166-f002:**
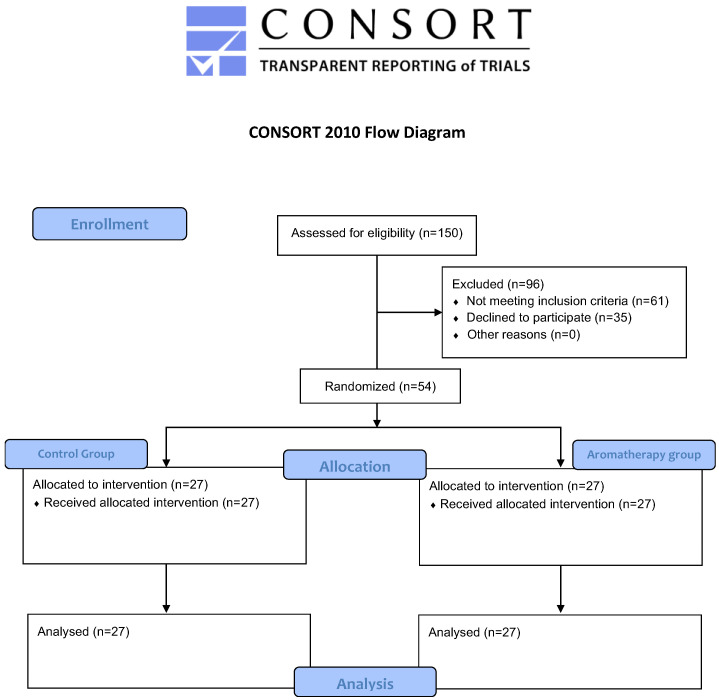
The flowchart of this study.

**Figure 3 medsci-13-00166-f003:**
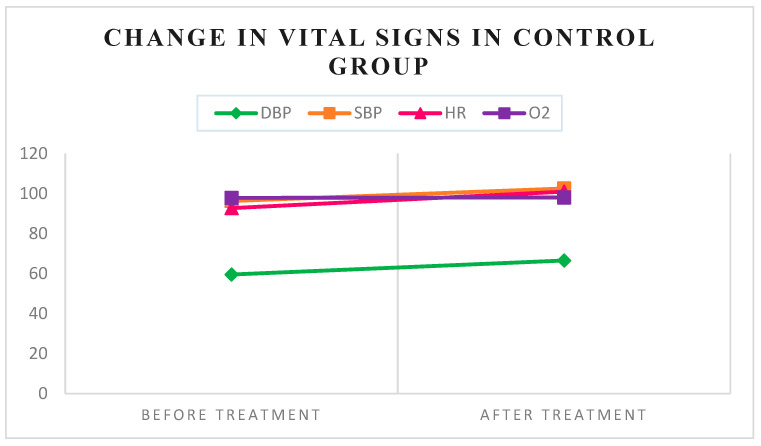
Changes in vital signs were observed in the control group.

**Figure 4 medsci-13-00166-f004:**
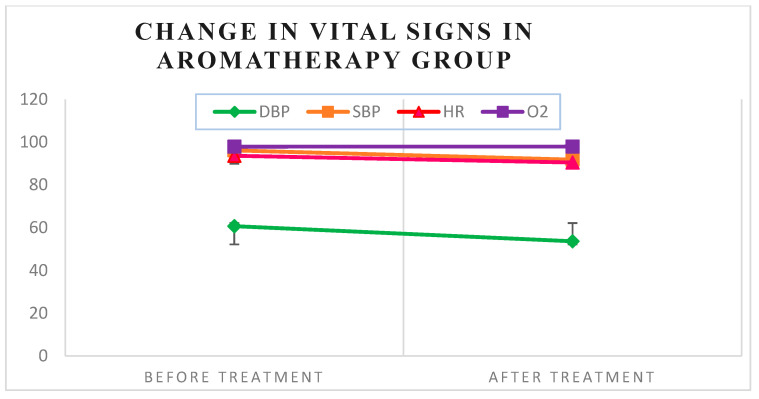
Changes in vital signs were observed in the aromatherapy group.

**Table 1 medsci-13-00166-t001:** Sample characteristics.

Groups	Age			Gender	
	Min	Max	Means ± SD	Boys (*n*%)	Girls (*n*%)
**Aromatherapy group**	6	11	8.1 ± 1.5	15 (44.44%)	12 (55.65%)
**Control group**	6	10	8.2 ± 1.3	12 (55.65%)	15 (44.44%)

**Table 2 medsci-13-00166-t002:** Within-group comparison.

	Variables	Before Treatment	After Treatment	After-Before Mean Differences ± SD	*p*-Value
**Control group**	FIS *	58.34	79.88		0.000 ^S^
	DBP ^$^	59.57 ± 11.59	66.61 ± 16.31	7.04 ± 15.57	0.024 ^S^
	SBP ^$^	96.36 ± 9.97	102.61 ± 11.33	6.25 ± 10.46	0.004 ^S^
	HR ^$^	92.79 ± 15.76	101.04 ± 19.43	8.25 ± 10.17	0.000 ^S^
	SPO2 ^$^	97.93 ± 1.09	98.11 ± 1.13	0.18 ± 1.33	0.485
**Aromatherapy group**	FIS *	55.11	51.7		0.6
	DBP ^$^	60.7 ± 12.07	53.67 ± 10.34	−7.04 ± 13.32	0.011 ^S^
	SBP ^$^	96.11 ± 13.46	91.81 ± 12.57	−4.30 ± 13.47	0.109
	HR ^$^	93.63 ± 16.83	90.48 ± 17.11	−3.15 ± 7.69	0.043 ^S^
	O2 ^$^	97.85 ± 1.61	97.93 ± 1.33	0.07 ± 1.44	0.791

*****: FIS Means rank Differences in value before and after treatment were analyzed using the Wilcoxon signed-rank test. ^$^: Means differences in value before and after treatment were analyzed using a paired *t*-test. ^S^: Significant differences were found within the group.

**Table 3 medsci-13-00166-t003:** Inter-group comparison.

	Before Anesthesia			After Anesthesia		
Variables	Control Group	Aromatherapy Group	*p*-Value	Control Group	Aromatherapy Group	*p*-Value
**FIS ***	58.34	55.11	0.916	79.88	51.7	0.001 ^S^
**DBP ^$^**	59.57 ± 11.59	60.7 ± 12.07	0.182	66.61 ± 16.31	53.67 ± 10.34	0.013 ^S^
**SBP ^$^**	96.36 ± 9.97	96.11 ± 13.46	1.00	102.61 ± 11.33	91.81 ± 12.57	0.038 ^S^
**HR ^$^**	92.79 ± 15.76	93.63 ± 16.83	0.825	101.04 ± 19.43	90.48 ± 17.11	0.000 ^S^
**SPO2 ^$^**	98.07 ± 1.24	97.85 ± 1.61	0.563	98.11 ± 1.13	97.93 ± 1.33	0.744
**FLACC ^$^**	-----	-----	-----	3.11 ± 2.42	1.51 ± 1.87	0.042 ^S^

FIS: Facial Image Scale, DBP: Diastolic blood pressure, SBP: Systolic blood pressure, HR: Heart rate, SPO_2_: O_2_ Oxygen saturation, FLACC: Face–Legs–Activity–Cry–Consolability. Mean ranks were used for the FIS, whereas means ± Standard Deviation (SD) were used for vital signs and the FLACC scale. *: Differences in mean ranks of FIS between groups were analyzed using the Mann–Whitney U test. ^$^: Differences in means of vital signs and FLACC scale between groups were analyzed using an independent T-test. ^S^: Significant differences between groups.

## Data Availability

The data that support the findings of this study are available from the corresponding author upon reasonable request.

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
