# Peer review of "Lavender–Neroli Aromatherapy for Reducing Dental Anxiety and Pain in Children During Anesthesia: A Two-Arm Randomized Controlled Trial"

_medsci, 2025, doi:10.3390/medsci13030166_

Round 1

Reviewer 1 Report

Comments and Suggestions for Authors

Dear Authors,

Below are some queries and suggestions for each section of your manuscript:

Abstract

 * Please specify the nature of the control in your study.

Introduction

 * This section is exceptionally well-written, structured, and comprehensive.

Materials & Methods

 * Please specify the institute and city where the Department of Pediatric Dentistry is located.

 * Kindly add the acronym for NSAIDs.

 * Lines 114-119 are not necessary. The time frame that needs to be specified in the Materials & Methods section should only pertain to patient recruitment and evaluation. The time spent on the ethical committee approval process and manuscript writing should not be mentioned. Please only include the time interval from patient recruitment to their final evaluation.

 * Paragraph 2.6: Please check the reference by Wajda et al.

Results

 * The results are clear, concise, and well-supported by explanatory tables and images.

Discussion

 * In the discussion, I would like the Authors to explore the potential future application of this method for children with special healthcare needs. For this patient group, other, mostly digital, distraction tools have been shown to be effective in reducing anxiety. Please refer to: "The Use of Audiovisual Distraction Tools in the Dental Setting for Pediatric Subjects with Special Healthcare Needs: A Review and Proposal of a Multi-Session Model for Behavioral Management. doi: 10.3390/children11091077." (This isn't my article, but I find it very interesting as it could offer some ideas for future perspectives for your paper, potentially enriching the discussion even further)

It would be highly interesting to add the perspectives of aromatherapy for this category of patients. Are there existing studies on this topic? What do they indicate? Given your current results in healthy children, what are the future prospects for special needs patients?

Author Response

Dear Esteemed Editor and Esteemed Reviewers,

We want to express our sincere appreciation for your valuable comments and constructive feedback on our manuscript entitled " Lavender-Neroli Aromatherapy for Reducing Dental Anxiety and Pain in Children During Anesthesia: A Two-Arm Randomized Controlled Trial."

We have carefully revised the manuscript in full compliance with your suggestions and recommendations. Below, we provide a detailed, point-by-point response addressing each comment.

Review 1:

Abstract

 * Please specify the nature of the control in your study.

Response: An overview of the control group details is added in the abstract.

Introduction

 * This section is exceptionally well-written, structured, and comprehensive.

Response: Thank you for your interest.

Materials & Methods

 * Please specify the institute and city where the Department of Pediatric Dentistry is located.

Response: This information about the institute and city has been added in the material and method (Study design) section.

* Kindly add the acronym for NSAIDs.

Response: The acronym (non-steroidal anti-inflammatory drugs) has been added.

* Lines 114-119 are not necessary. The time frame that needs to be specified in the Materials & Methods section should only pertain to patient recruitment and evaluation. The time spent on the ethical committee approval process and manuscript writing should not be mentioned. Please only include the time interval from patient recruitment to their final evaluation.

Response: these lines were removed, the time interval from the patient recruitment to their final evaluation exists in the intervention procedures section.

 * Paragraph 2.6: Please check the reference by Wajda et al.

Response: The reference is checked and has been removed, and the explanation about the choice of 5 minutes before the intervention and during it is added in the discussion section.

Results

 * The results are clear, concise, and well-supported by explanatory tables and images.

Response: Thanks for your interest.

Discussion

 * In the discussion, I would like the Authors to explore the potential future application of this method for children with special healthcare needs. For this patient group, other, primarily digital, distraction tools have been shown to be effective in reducing anxiety. Please refer to: "The Use of Audiovisual Distraction Tools in the Dental Setting for Pediatric Subjects with Special Healthcare Needs: A Review and Proposal of a Multi-Session Model for Behavioral Management. doi: 10.3390/children11091077." (This isn't my article, but I find it very interesting as it could offer some ideas for future perspectives for your paper, potentially enriching the discussion even further)

It would be exciting to incorporate aromatherapy perspectives into this category of patient care. Are there existing studies on this topic? What do they indicate? Given your current results in healthy children, what are the prospects for special needs patients?

Response: in the fact one of the suggestion is to study the effect of aromatherapy in the children with special need, aromatherapy deal with sense of smell, people may can refuse to see or hear but they can’t prevent themself from smell odors, which make it promising approach in this group of children as it can be apply with the less of corporation such as which found in this group of children, essential oils molecules can pass through the olfactory bulb, and reach to the brain when they can do there anxiolytic effect, its right that audiovisual distraction can help reduce anxiety in children with special need, but some of them such VR eyeglass maybe seem frightened and lead to less corporation as happened with down syndrome children in the studies mentioned above, such technique may isolate child from the environment which evoke anxiety in those children, while aromatherapy can diffuse in many methods, with the lack of studies in this area it can be an promising topic for further research.

Revised text in the discussion section: Given the promising anxiolytic effects observed in this study, future research could explore the application of aromatherapy for children with special healthcare needs, where conventional audiovisual distraction tools may not always be feasible or practical. Since olfactory stimuli do not require active cooperation and are less likely to induce sensory overload, aromatherapy may offer a gentle, non-invasive alternative for anxiety management in this vulnerable population, warranting further investigation.

Reviewer 2 Report

Comments and Suggestions for Authors

The manuscript is easy to read and adds value to the existing literature. By conducting a randomized controlled trial, the authors demonstrated that breathing in essential oils for 5 minutes before ID block can be beneficial for pediatric patients compared to breathing in normal air.

My main concern is that the modified nasal nitrous oxide mask may be frightening to children. Children may not want to be face-covered by a mask, which may symbolize containment, illness, and other thoughts. With essential oil fragrance, the intervention group children may be more tolerable to the mask. Without the fragrance, the control group children may be more scared and anxious than they normally are, due to being masked for no obvious reason (from their perspective).

Other than that, other components reported in this manuscript sound reasonable and valid.

Author Response

Reviewer 2:

The manuscript is easy to read and adds value to the existing literature. By conducting a randomized controlled trial, the authors demonstrated that breathing in essential oils for 5 minutes before ID block can be beneficial for pediatric patients compared to breathing in normal air.

My primary concern is that the modified nasal nitrous oxide mask may be frightening to children. Children may not want to be face-covered by a mask, which may symbolize containment, illness, and other thoughts. With essential oil fragrance, the intervention group children may be more tolerable to the mask. Without the fragrance, the control group children may be more scared and anxious than they normally are, due to being masked for no obvious reason (from their perspective).

Other than that, other components reported in this manuscript sound reasonable and valid.

Response:

I appreciate your interest; however, some points need clarification. First, Nitrous oxide mask is covered only the child's nose allowing them to see still and feel their environment, the issue arises when using the usual nitrous oxide mask, which prevent the air from passing into child nose, so until pumping oxygen the child can feel difficulty in breathing which make him frightened, in this study the mask was ventilate which allow the air exchange, also it had that cute design with the box on it, it look like the nose of a cat or a bear most of children were happy to used it as they think about it as a toy, second, the child before putting the mask didn’t know about the odor of the essential oils, so the decision to put the mask didn’t depends on the desire to smell an odor, it is right that it could be something new if an control odor is used in the control group, to compare the effect of the odor in compare the impact of the essential oils molecules, it was mentioned in the limitation section.

Reviewer 3 Report

Comments and Suggestions for Authors

Thank you for the  article titled "Lavender-Neroli Aromatherapy for Reducing Dental Anxiety and Pain in Children During Anesthesia: A Two-Arm Randomized Controlled Trial." The study addresses a significant clinical issue and contributes interesting insights into the development of non-invasive methods for supporting treatment in pediatric dentistry, but in my opinion, the authors should improve the manuscript.

Introduction

  • The authors provide a strong introduction to the study; however, a brief discussion of existing non-pharmacological methods for reducing anxiety in children, such as hypnosis, relaxation techniques, and virtual reality, would be beneficial. This addition would place aromatherapy within a broader clinical context.
  • Additionally, discussing the safety of using essential oils in children, including potential adverse effects, would strengthen the rationale for choosing this method to prepare children for the procedure.

Study Objective (Lines 83–85)

The objective is clearly stated. It would be helpful to incorporate a sentence about the research hypothesis.

Materials and Methods

  • Lines 159–164: The authors provide a detailed description of the method used to prepare the oil mixture and its application via a modified mask. It would be useful to mention whether the use of the 3D-printed box/mask has previously been validated for airflow and safety.
  • Additionally, data on how fragrance intensity was assessed is lacking—specifically, whether it was subjectively evaluated by the children or objectively controlled by the researchers.

The results section is factually accurate and clearly presents the main findings. I recommend expanding this section to include:

  • A more detailed presentation of the demographic characteristics of the study groups.
  • Maybe an analysis of the impact of variables such as age, gender, and previous dental experience through subgroup analysis.

Discussion

The authors demonstrate a good understanding of the literature and acknowledge the study's limitations. Maybe additional comments may include:

  • Although aromatherapy is generally considered safe, it is not without risks, especially for children with undiagnosed bronchial hyperreactivity or allergies.
  • The potential impact of gender, age, and previous dental experience should be considered—conducting a subgroup analysis could enhance the interpretation of the results.

Author Response

Reviewer 3

Thank you for the article titled "Lavender-Neroli Aromatherapy for Reducing Dental Anxiety and Pain in Children During Anesthesia: A Two-Arm Randomized Controlled Trial." The study addresses a significant clinical issue and contributes interesting insights into the development of non-invasive methods for supporting treatment in pediatric dentistry, but in my opinion, the authors should improve the manuscript.

 Introduction

  • The authors provide a strong introduction to the study; however, a brief discussion of existing non-pharmacological methods for reducing anxiety in children, such as hypnosis, relaxation techniques, and virtual reality, would be beneficial. This addition would place aromatherapy within a broader clinical context.

Thank you for your mention of this idea. A discussion on the recent literature regarding new non-pharmacological methods has been added.

  • Additionally, discussing the safety of using essential oils in children, including potential adverse effects, would strengthen the rationale for choosing this method to prepare children for the procedure.

Response: Thank for your attention to this part, when consideration the use of complementary medicine products such as essential oils a wide range of safety is the first thing that comes to mind, however it’s not that simple, although aromatherapy it can consider safe but it can have some adverse effects especially for individuals who are sensitive to essential oils or when oils are applied direct to the skin undiluted, which can irritate, this study is the only one that concerned with complications in using aromatherapy in pediatric dental treatment, that mentioned in the discussion section where a new methods is development to deliver aromatherapy and a dilution for is the essential oils is conducted, all this to enhance the safety of aromatherapy practices.

Study Objective (Lines 83–85)

The objective is clearly stated. It would be helpful to incorporate a sentence about the research hypothesis.

Response: Thank you for your interest. The research hypothesis may be explained well through the research purpose, which includes the variables that are used in this article to assess the efficacy of aromatherapy with lavender-neroli oil blend in the management of dental anxiety.

Materials and Methods

  • Lines 159–164: The authors provide a detailed description of the method used to prepare the oil mixture and its application via a modified mask. It would be helpful to mention whether the use of the 3D-printed box/mask has previously been validated for airflow and safety.

Response: Thanks for highlighting this important aspect. In the initial design of the box, it was only a cylindrical box with one chamber perforated from its upper and bottom, but the cotton balls could block the holes inside them and hinder ventilation. Therefore, this design was resorted to, which includes four chambers. These chambers are separated from each other and perforated from the bottom, as shown in the attached pictures. The cotton balls were placed only in three chambers, leaving one empty to ensure continued ventilation.

  • Additionally, data on how fragrance intensity was assessed is lacking—specifically, whether it was subjectively evaluated by the children or objectively controlled by the researchers.

Response:

During the selection of mixture concentrations, several options were considered, and 20 children were asked for their preferences, leading to the final choice of concentrations. This data has been added in the discussion section.

The results section is factually accurate and presents the main findings. I recommend expanding this section to include:

  • A more detailed presentation of the demographic characteristics of the study groups.
  • Maybe an analysis of the impact of variables such as age, gender, and previous dental experience through subgroup analysis.

Response: A table has been added showing the differences in sample characteristics in terms of age and gender.

Ghaderi etal and Pradopo etal found no differences between boys and girls when aromatherapy is used to reduce anxiety. This could be due to the small sample size for that type of research, which studies the effect of gender and age on the results of aromatherapy on anxiety.

It could be a suggestion for further studies using a larger sample size and a broader age range. The current research examines the efficacy of aromatherapy in only the school children in the age group, which limits the realism of the results. By including a more diverse group, we can obtain more accurate results about the effect of age and gender in the aromatherapy.

Discussion

The authors demonstrate a good understanding of the literature and acknowledge the study's limitations. Maybe additional comments may include:

  • Although aromatherapy is generally considered safe, it is not without risks, especially for children with undiagnosed bronchial hyperreactivity or allergies.
  • The potential impact of gender, age, and previous dental experience should be considered—conducting a subgroup analysis could enhance the interpretation of the results.

Response:

I appreciate your interest. It is essential to highlight that this is the only study that examined the safety of aromatherapy for managing dental anxiety in children. In this study, children with respiratory problems were included. The aromatherapy was provided through a mask connected to a separate box that contained the allergen, allowing for its removal if necessary. Additionally, oxygen was supplied in the event of an allergic reaction.

Reviewer 4 Report

Comments and Suggestions for Authors
  • Suggestion: Please include the intervention method for the control group and the statistical analysis method in the abstract.

  • As mentioned in the introduction of this study, there are studies that have separately investigated lavender or sweet orange oils, or lavender with neroli oil used as a blend in combination with music therapy. This study uses lavender with neroli oil, which has minimal innovation.

  • The sample size estimation was based on changes in pulse rate values, but dental anxiety and pain scores were defined as the primary outcome measures. Ideally, the sample size should have been estimated using the primary outcome measures, so this is problematic.

Author Response

Reviewer 4:

Suggestion: Please include the intervention method for the control group and the statistical analysis method in the abstract.

Response: This data has been added to the abstract.

As mentioned in the introduction of this study, some studies have separately investigated lavender or sweet orange oils, or lavender with neroli oil used as a blend in combination with music therapy. This study uses lavender with neroli oil, which has minimal innovation.

Response: Literary development involves small steps that accumulate to create significant changes. The innovation in this study did not include the choice of the nature of the oils, but rather the concept of mixing for inhalation, which might contradict the intended results. In addition to the development in the technique of applying aromatherapy and the interest in the issue of safety in the use of essential oils, which no other study has addressed, in addition to that, many shortcomings in previous studies were addressed, such as the failure to mention the analysis of the components of the essential oils used, in this study the element of the using essential oils were determined by gas chromatography which makes their results more reliable. This study can contribute significantly to the advancement of aromatherapy for managing dental anxiety in children.

The sample size estimation was based on changes in pulse rate values, but dental anxiety and pain scores were defined as the primary outcome measures. Ideally, the sample size should have been estimated using the primary outcome measures, so this is problematic.

Response: The sample size was calculated based on heart rate changes observed in a similar study, which may not provide sufficient power to detect statistically significant differences in the other primary variables assessed. Therefore, findings related to those additional variables should be interpreted as exploratory for future research.

Round 2

Reviewer 2 Report

Comments and Suggestions for Authors

The authors replied that the the mask "look like the nose of a cat or a bear most of children were happy to used it as they think about it as a toy." But the mask from Figure 1C does not look like that. The authors can show a picture of the mask from the front, and report data that show how happy the children were to use it.

Author Response

The authors replied that the the mask "look like the nose of a cat or a bear most of children were happy to used it as they think about it as a toy." But the mask from Figure 1C does not look like that. The authors can show a picture of the mask from the front, and report data that show how happy the children were to use it.

Response: This text has been added (in pink) to explain the meaning and address the reviewer's note. 

Following its modification, the nasal mask was reframed and introduced to children not as a clinical device but as a playful object. The mask, introduced to resemble familiar figures such as a cat or bear, created an appearance that aligned more closely with a toy than with medical equipment. This design adaptation, coupled with the use of simplified, child-friendly terminology, contributed to reducing anxiety and enhancing cooperation during dental procedures. By integrating elements of play and imagination, the intervention improved children's acceptance of care and facilitated their understanding of the procedures being performed. Such approaches underscore the value of child-centered communication and environmental adaptation in pediatric dentistry, where minimizing fear can significantly influence treatment outcomes and long-term attitudes toward oral healthcare.

Reviewer 4 Report

Comments and Suggestions for Authors

The authors have addressed my concerns.

Author Response

The authors have addressed my concerns.

Response: Thank you.